# Current Standards in the Management of Early and Locally Advanced Cervical Cancer: Update on the Benefit of Neoadjuvant/Adjuvant Strategies

**DOI:** 10.3390/cancers14102449

**Published:** 2022-05-16

**Authors:** Yuedan Zhou, Elie Rassy, Alexandre Coutte, Samir Achkar, Sophie Espenel, Catherine Genestie, Patricia Pautier, Philippe Morice, Sébastien Gouy, Cyrus Chargari

**Affiliations:** 1Department of Radiation Oncology, Centre Hospitalier Universitaire, 80000 Amiens-Picardie, France; zhou.yuedan@chu-amiens.fr (Y.Z.); coutte.alexandre@chu-amiens.fr (A.C.); 2Department of Medical Oncology, Gustave Roussy Comprehensive Cancer Centre, 94800 Villejuif, France; elie.rassy@gustaveroussy.fr (E.R.); patricia.pautier@gustaveroussy.fr (P.P.); 3Department of Radiation Oncology, Gustave Roussy Comprehensive Cancer Centre, 94800 Villejuif, France; samir.achkar@gustaveroussy.fr (S.A.); sophie.espenel@gustaveroussy.fr (S.E.); 4Department of Pathology, Gustave Roussy Comprehensive Cancer Center, 94800 Villejuif, France; catherine.genestie@gustaveroussy.fr; 5Department of Surgery, Gustave Roussy Comprehensive Cancer Center, 94800 Villejuif, France; philippe.morice@gustaveroussy.fr (P.M.); sebastien.gouy@gustaveroussy.fr (S.G.)

**Keywords:** cervical cancer, radiation oncology, brachytherapy, chemotherapy, immunotherapy

## Abstract

**Simple Summary:**

Cervical cancers is a human papillomavirus infection-induced gynecologic cancer. Due to the uneven access to prevention measures in the world, it is still a leading cause of cancer death in women in low- and middle-income countries. The mainstay of treatment for early-stage cervical cancers is upfront surgery. Clinical trials confirmed the place of adjuvant radiotherapy to improve disease control, but also highlighted the need for a careful selection of patients prior to surgery, in order to avoid the cumulative morbidities of each treatment. In locally advanced cervical cancers, the standard of care remains concurrent pelvic chemoradiotherapy followed by an image-guided adaptive brachytherapy boost allowing for dose escalation and leading to a very high probability of local control. Systemic failures remain a major concern, and neoadjuvant or adjuvant approaches in this context are discussed in the light of recent literature.

**Abstract:**

Globally, cervical cancers continue to be one of the leading causes of cancer-related deaths. The primary treatment of patients with early-stage disease includes surgery or radiation therapy with or without chemotherapy. The main challenge in treating these patients is to maintain a curative approach and limit treatment-related morbidity. Traditionally, inoperable patients are treated with radiation therapy solely and operable patients undergo upfront surgery followed by adjuvant (chemo) radiotherapy in cases with poor histopathological prognostic features. Patients with locally advanced cervical cancers are treated with concurrent chemoradiotherapy followed by an image-guided brachytherapy boost. In these patients, the main pattern of failure is distant relapse, encouraging intensification of systemic treatments to improve disease control. Ongoing trials are evaluating immunotherapy in locally advanced tumours following its encouraging efficacy reported in the recurrent and metastatic settings. In this article, clinical evidence of neoadjuvant and adjuvant treatments in cervical cancer patients is reviewed, with a focus on potential strategies to improve patients’ outcome and minimize treatment-related morbidity.

## 1. Introduction 

Cervical cancer is mainly caused by high-risk oncogenic human papillomavirus, a common sexually transmitted infection of the lower genital tract. The World Health Organization encourages the implementation of primary and secondary prevention measures (vaccination and screening, respectively) that would favor the eradication of the disease [1]. The backdrop for this intervention is based on the epidemiology of cervical cancer which currently ranks as the fourth most frequently diagnosed cancer (604,000 new cases), and the fourth leading cause of cancer death in women (342,000 deaths) worldwide in 2020 [2]. Approximately 90% of new cases and deaths occur in low- and middle-income countries, where it is the third most common cancer among women. This epidemiologic distribution has important clinical implications, with patients being frequently diagnosed at an advanced stage of their disease, in contexts where effective treatments are not available in all countries in the world [3]. The diagnosis of cervical cancer is traditionally made by histologic evaluation of a biopsy of the primary or metastatic locations; however, the staging evaluation varies according to the available resources. The staging workup minimally encloses a detailed physical examination and endoscopic diagnostic procedures (examination under anesthesia, proctoscopy, cystoscopy, hysteroscopy) are discussed according to each specific situation. If possible, the diagnostic evaluation includes at least a computed tomography, or ideally a combination of magnetic resonance imaging (MRI) and a 18-fluorodeoxyglyucose positron emission tomography (18-FDG PET/CT) that are recommended as part of a modern therapeutic management [4,5,6]. 

Cervical cancer spreads by direct extension into the vagina, parametrium, uterine corpus, and adjacent organs; it spreads along the lymphatic channels to the regional (obturator, external iliac, internal iliac) and distant (common iliac and para-aortic nodes) lymph nodes, and metastasizes to the lungs, liver, and skeleton via the hematogenous route [7]. In 2018, the Fédération Internationale de Gynécologie et d’Obstétrique (FIGO) committee published a staging system that considers clinical, radiological, or pathological findings, as available, to assign the stage [8]. The primary treatment options for patients with invasive cervical cancer include radical hysterectomy or fertility-sparing surgery (cone resection or trachelectomy) in highly selected patients, or radiation therapy with or without chemotherapy and followed with a brachytherapy boost [9,10]. Patients with early-stage cervical cancers treated with upfront surgery undergo adjuvant radiotherapy +/− chemotherapy in cases of adverse histopathological factors. Patients with locally advanced cervical cancer (LACC) are treated with chemoradiation followed by an image-guided adaptive brachytherapy boost. These strategies yield a satisfactory local control rate (>90–95%), but the occurrence of distant metastases underlines the necessity for a systemic therapeutic strategy in order to decrease the risk of distant failure [10]. Here, we review the clinical evidences for neoadjuvant/adjuvant treatments in cervical cancers and highlight strategies to improve patient outcome. The contexts of fertility sparing approaches and of cervical cancer occurring in pregnancy are excluded from this literature review, as both issues were recently addressed extensively [11,12].

## 2. Early-Stage Cervical Cancer 

Early-stage cervical cancers refer to tumors ≤ 4 cm in the largest dimension, and are restricted to the uterine cervix without lymph node involvement, thus enclosing the FIGO 2018 stage IA, IB1, and IB2 diseases.

### 2.1. Upfront Surgical Treatment

Upfront radical hysterectomy is suitable for early-stage invasive cervical cancers, with a degree of parametrial resection that depends on histopathological prognostic factors (depth of infiltration, presence of lymphovascular involvement). The surgery of the uterus is associated with pelvic lymph node dissection (PLND), including external iliac, inter-arteriovenous, and obturator lymph nodes [8,13,14,15]. The procedure of a sentinel lymph node (SLN) biopsy is being increasingly used, for ultra-staging but also, given the significant morbidities reported in PLND, such as hemorrhaging, ureteral and/or nerve lesions, as well as lymphoedema [16,17]. The surgery of the uterus is recommended after ruling out, intraoperatively, the pelvic lymph node extension. The optimal surgical approach remains debatable. A randomized study showed that minimally invasive surgery was associated with more frequent disease relapses, compared to laparotomy-based hysterectomies (3-year disease-free survival of 91.2% vs. 97.1%, HR 3.74, 95% CI, 1.63 to 8.58) [18]. In a large retrospective cohort study, the deleterious impact of mini-invasive surgery was observed among the subgroup of patients with a tumor size > 2 cm (HR 2.31; 95% CI 1.37–3.90) [19]. 

### 2.2. Indications for Adjuvant (Chemo)radiotherapy

Around 25–40% of stage IB cervical cancers present risk factors identified on the hysterectomy pathology specimen and required adjuvant treatment [20]. Patients with one or more adverse histopathological factors, such as positive or close surgical margins, positive lymph nodes, or microscopic parametrial involvement are at a high risk of relapse. Several retrospective series studied the long-term results of patients operated for early-stage cervical cancers without adjuvant treatments. 

The most significant prognostic factors for stage IB cervical cancer are occult pelvic lymph node involvement found at time of surgery, and tumor size [21,22,23]. After a radical hysterectomy, the 5-year OS rate for patients with tumors ≤ 2 cm was 94–97%, compared with 69.9–80% of patients with tumors > 4 cm [22]. One randomized phase III study compared adjuvant chemoradiotherapy versus adjuvant pelvic radiotherapy in 268 patients. They had FIGO IA2-IIA cervical cancer, with positive pelvic lymph nodes and/or positive margins, and/or microscopic involvement of parametrium after an upfront radical hysterectomy and PLND [24]. Chemotherapy consisted of 3 cycles of cisplatin, plus fluorouracil every 3 weeks, with the first and second cycles given concurrently with the radiotherapy, which is 49.3 Gy in 29 fractions to the pelvic field. The 4-year progression-free survival (PFS) and 4-year OS were both significantly improved in the adjuvant chemoradiotherapy group compared with the adjuvant radiotherapy group (80% vs. 63% and 81% vs. 71%, respectively). The addition of concurrent chemotherapy to adjuvant radiotherapy is therefore considered to be the standard to improve survival of high-risk, early-stage cervical cancer patients [9]. 

Along with the abovementioned high-risk factors, tumor dimensions, deep stromal invasion, and lymphovascular space invasion, are intermediate-risk factors increasing the risk of relapse when combined [9,25]. A phase III randomized Gynecologic Oncology Group (GOG) trial including 277 patients with at least two intermediate-risk factors, compared adjuvant pelvic radiotherapy with a radical hysterectomy plus PLND alone. The planned pelvic dose was from 46 Gy in 23 fractions to 50.4 Gy in 28 fractions. A significant increase in 10-year PFS was observed in the group of patients who received post-operative radiotherapy, but the 10-year OS were similar in two groups [26,27]. This randomized clinical trial showed the benefits of adjuvant pelvic radiotherapy in these patients; however, MRI was not routinely used in the 1990s to evaluate the disease extension in the pelvis, and 18-fluorodeoxyglucose positron emission tomography/computed tomography (18-FDG PET/CT) only came into regular use a decade later. The disease staging is much more precise with modern imaging methods, possibly leading to fewer indications for adjuvant treatment through a better selection of patients for upfront surgery [4,6]. In the GOG trial published by Sedlis et al. in 1999, the criteria for adjuvant radiotherapy were defined for squamous cell carcinoma at an estimated recurrence risk of at least 30%, which is a very high threshold [24]. More recently, an ancillary analysis of GOG 49, 92, and 141 trials was conducted to better identify the risk of relapse according to tumor histopathological characteristics. Authors proposed histology specific nomograms that better represented the risk of relapse to guide adjuvant radiotherapy. Though the optimal threshold to indicate adjuvant radiotherapy is still unclear, they showed that risk factors for recurrence differed for squamous cell carcinoma and adenocarcinoma of the cervix, with tumor size being the risk factor associated with the highest risk for relapse in adenocarcinoma, and for squamous cell carcinoma, depth of invasion is the most important risk factor [25]. The place of adjuvant chemoradiation in patients with a combination of intermediate-risk factors is still under investigation.

Brachytherapy boost is frequently added in the adjuvant setting to the external radiotherapy to treat the vaginal cuff, where most locoregional relapses occur in patients treated with primary surgery. The benefit of adding vaginal brachytherapy was not clearly demonstrated [28], but retrospective series suggested that a treatment combining adjuvant intensity-modulated radiation therapy (IMRT) and a brachytherapy boost after radical hysterectomies in patients with cervical cancer, was associated with excellent long-term outcomes and limited rates of non-hematologic toxicity in patients with FIGO IB1-IIIC2 cervical cancers [29]. A retrospective study compared the treatment effect in 480 patients with at least two intermediate-risk factors, who received external beam radiotherapy with or without vaginal brachytherapy [28]. Patients who received brachytherapy had worse prognostic factors. The 5-year OS, 5-year local recurrence-free survival, and 5-year distant metastasis-free survival were all similar in both groups. Scarce data supported the use of vaginal vault brachytherapy in patients with positive surgical margins [28]. In selected situations, adjuvant vaginal brachytherapy was used as an exclusive adjuvant treatment in patients with pT1a-1b, pN0 cervical cancer and intermediate histopathological risk factors (LVSI) after a radical colpohysterectomy, in order to avoid the morbidity of adjuvant EBRT. In the only published series testing exclusive adjuvant brachytherapy, 10% of patients experienced relapse all having peritoneal cavity or lymph node failure. No vaginal or isolated pelvic nodal failure [30]. 

### 2.3. Upfront Radiotherapy to Avoid Cumulative Morbidities

In a randomized study including patients with stage IB and IIA cervical carcinoma, radical surgery and radiotherapy showed similar survival outcomes, but patients in the surgery arm had a higher proportion of treatment-related adverse events [31,32]. Two-thirds of the patients in the radical surgery arm required adjuvant radiotherapies due to pejorative prognostic factors (stage T2b or greater disease, less than 3 mm of uninvolved cervical stroma, positive margin and lymph node metastases). In approximately one half of the cases, the indication was based on local risk factors, such as deep stromal invasion, positive margins, parametrial extension. Adjuvant radiotherapy increased the risk of post-treatment complications. One major difficulty of post-operative radiotherapy in patients who had their uterus removed is that gastrointestinal structures, especially the bowel, are more affected than in upfront radiotherapy [31]; therefore, patients with tumors that are close to 35–40 mm, or with pejorative factors identified from conization, may be preferentially treated with upfront chemoradiation plus brachytherapy. When there are indications that postoperative radiotherapy will be necessary, chemoradiation plus brachytherapy should be preferred to avoid cumulative morbidities of surgery and adjuvant treatment. Radiotherapy is also indicated for treatment of patients with medical contraindication for surgery, followed by a brachytherapy boost. 

### 2.4. Place of Image-Guided Intensity-Modulated Radiotherapy (IG-IMRT)

Adjuvant postoperative and upfront radiotherapy cause long-term gastrointestinal tract symptoms in patients treated with pelvis radiotherapy or brachytherapy for gynecological malignancies; the risk of long-term severe gastrointestinal sequelae still persists for a long time after radiotherapy [24,26,31,33,34,35,36,37,38]. Two phase II studies demonstrated that post-operative IMRT to the pelvis in endometrial carcinomas caused an acceptable rate of short-term bowel adverse events and gastro-intestinal toxicities [39,40]. The randomized phase III NRG 1203 trial including 278 patients with gynecologic malignancies compared patient-reported acute toxicity and quality of life using a standard outcome questionnaire after the post-operative standard pelvic 3D-CRT and IMRT. Pelvic IMRT was associated with significantly less gastrointestinal and urinary tract toxicities compared to 3D-CRT at week 5 and 1 year [41]. A recent randomized phase III PARCER trial including 300 patients with cervical cancers, also compared late gastrointestinal tract toxicities after post-operative IMRT with daily image-guidance compared with 3D-CRT. Seventy-seven percent of the patients also received concurrent chemotherapy. IG-IMRT was associated with a significantly lower 3-year incidence of grade ≥ 2 late gastrointestinal tract toxicities compared with 3D-CRT (21.1% vs. 42.4%, HR, 0.46; 95% CI, 0.29 to 0.73; *p* < 0.001), as well as a 3-year incidence of grade ≥ 2 with any late toxicities (28.1% vs. 48.9% HR, 0.50; 95% CI, 0.33 to 0.76; *p* < 0.001). No difference in the 3-year disease control was observed, confirming the safety of IMRT technique [42]. Evidences from these two randomized phase III studies established the superiority of IMRT as adjuvant radiotherapy in gynecological cancers to decrease treatment-related morbidities, without jeopardizing treatment efficacy. 

### 2.5. Neoadjuvant Brachytherapy

In order to reduce the treatment-related risk of combining surgery and adjuvant radiotherapy, neo-adjuvant brachytherapy in early-stage is propsoed in a few expert centers to patients with unfavorable histoprognostic factors. Retrospective series have reported on the efficiency and safety of neo-adjuvant brachytherapy, which consists of performing intracavitary brachytherapy 6 to 8 weeks before surgery. Rretrospective studies including patients with squamous cell carcinoma and adenocarcinoma suggested that neo-adjuvant brachytherapy is an attractive option for patients with aforementioned local pejorative prognostic factors such as lympho-vascular invasion or a tumor size > 2 cm. Neo-adjuvant brachytherapy is a valid option according to the latest European guidelines for cervical cancer management in centers familiar with this approach [9,43]. The brachytherapy delivers a considerable dose of radiation to the cervix uteri, the proximal part of the parametrium, and upper third of the vagina [43]. Due to the very sharp dose gradient of brachytherapy, a dose received by at risk organs in the vicinity, especially the bowel, is usually low. In retrospective data, 70% of patients have achieved a complete histological response by the time of surgery [44]. The 5-year estimated OS was 84.5%, and the 5-year DFS was 84.4%, with minimal treatment-related complications. In addition, fewer than 1% of patients have residual tumor cells in the parametrium, suggesting that extra-fascial hysterectomy could be an alternative to colpohysterectomy in this context [44]. A retrospective study compared upfront surgery with adjuvant radiation therapy (external beam radiation and/or vaginal brachytherapy) or neo-adjuvant radiotherapy (mainly uterovaginal brachytherapy) followed by surgery. This series showed that more frequent post-operative acute ureteral complications were observed in patients with neo-adjuvant radiotherapy (2.3% vs. 0.6%), but 10-year grade 3 and 4 late treatment-related complications were three times more frequent among patients treated with adjuvant external radiation compared with patients treated with neo-adjuvant brachytherapy (22% vs. 7%) [45]. 

### 2.6. Challenging Situations 

With the development of the sentinel lymph node (SLN) biopsy [7], and pathological ultra-staging of SLN techniques, new questions have emerged, including the strategy to adopt the adjuvant treatment when micrometastases (lymph node metastases > 0.2 mm and up to 2 mm) or isolated tumor cells (ITCs) (tumor cells clusters < 0.2 mm) emerge. By doing multiple serial sectioning with immunohistochemistry staining, pathologic ultra-staging increases the detection rate of low-volume disease [46]. In 4–15% of SLN biopsy samples from early-stage cervical cancer patients, micrometastases were identified [47,48]. An international retrospective study examined the significance of micrometastases and isolated tumor cells for the disease prognosis in 645 patients treated for FIGO stage IA-IIB cervical cancers [49]. Macrometastases, micrometastases, and isolated tumor cells were detected by ultra-staging combined with pelvic non-SLN in 21.1%, 7.1%, and 3.9% of patients, respectively [49]. In this study, 85.3% of patients with macrometastases, 82.6% with micrometastases, 52% with isolated tumor cells, and 10.5% with negative pelvic nodes, received adjuvant therapy. The presence of isolated tumor cells was not a prognostic factor for recurrence free survival or overall survival; however, the presence of macrometastases or micrometastases was significantly associated with reduced overall survival, which was comparable to that of patients with macrometastases [49]. The impact of micrometastases and isolated tumor cells on survival was also examined in SLN samples from 139 cervical cancer patients with FIGO IA2-IB1 tumors treated in the SENTICOL I study [50]. In this study, the presence of the micrometastatses was not a prognostic factor for disease-free survival, possibly due to the lack of statistical power [50]. The adaptation of adjuvant treatment strategies should probably consider the presence of micrometastases, especially for indication of adjuvant radiotherapy, but not the presence of isolated tumor cells in the SLN biopsy.

### 2.7. Sequential Adjuvant Chemoradiation

There is currently no consensus concerning the role of sequential chemotherapy in addition to postoperative radiation in patients with early-stage cervical cancer who show adverse pathological factors. The sequence of chemotherapy using paclitaxel–carboplatin every 3 weeks, followed by radiotherapy, did not improve 2-year PFS or 5-year OS compared with the concomitant chemoradiotherapy using weekly cisplatin (81.8% and 87.2%, *p* = 0.235) in a phase III randomized trial including 263 patients; however, the sequential chemotherapy had lower hematological toxicities, but a much higher neurotoxicity and alopecia rate [51]. The recent phase III STARS clinical trial, including 1048 patients with FIGO IB to IIA tumors, showed that adjuvant sequential chemoradiotherapy using paclitaxel–cisplatin every 3 weeks was associated with a higher rate of 3 year DFS compared with radiotherapy alone (90% vs. 82%, HR, 0.52, 95% CI 0.35–0.76), or to concomitant chemoradiotherapy (90% vs. 85%, HR, 0.65, 95% CI 0.44–0.96), and a higher 5 year OS compared with adjuvant radiotherapy alone (92% vs. 88%, HR, 0.58; 95% CI 0.35–0.95) [52]. This study had some limitations, but it suggests a potential benefit of adjuvant sequential chemoradiation in patients with high-risk features. 

## 3. Locally Advanced Cervical Cancer

According to the FIGO, LACC refers to tumors ≥ 4 cm in its largest dimension, or extending outside the cervix (e.g., with vaginal invasion, parametrial involvement, or extension to proximity organs) or with lymph node metastases [6]. 

### 3.1. Standard of Care

Patients with LACC are treated according to a well-defined strategy relying on concurrent chemoradiotherapy with 5 cycles of weekly cisplatin 40 mg/m^2^ followed by a sequential brachytherapy boost [24,53,54,55,56,57,58]. The benefit of chemoradiation compared with radiotherapy alone was a 6% improvement in 5-year OS and an 8% improvement in 5-year DFS through improvements in local control and reductions in distant failures [59]. Dose intensity is a major factor for disease control. The radiotherapy consists of 45–46 Gy, delivered in 25 fractions to the pelvis +/− para-aortic lymph nodes according to primary lymph node staging, with an external radiotherapy boost to macroscopic lymph nodes [43,60,61,62,63,64,65]. Surgical para-aortic lymph node staging can be discussed to guide radiotherapy volumes in patients with pelvic lymph node metastases, but the clinical benefit of this strategy was not demonstrated [61,62]. There is no demonstrated benefit of performing a debulking of enlarged pelvic lymph nodes. Indeed, chemoradiotherapy with an additional external beam radiotherapy boost is usually sufficient to achieve high lymph node control probability and the surgical lymph node removal increases the risk of lower limb edema [66]. Brachytherapy is part of the standard of care for LACC to focally increase the dose to the cervix uteri and potential residual disease [57]. It is associated with a decrease in local relapses and severe toxicities, as well as being a major benefit for survival [67,68,69,70,71,72]. Integration of 3D treatment planning brachytherapy based on MRI evaluation further improved local control and decreased toxicities by delivering higher doses to the tumor and minimizing doses in organs at risk [43,60,67,72]. The overall treatment time is also crucial for treatment efficacy, with retrospective clinical evidence noting that overall treatment time, from the first radiotherapy session to brachytherapy completion, should be ideally less than 51–52 days to maximize local control [70,73,74,75]. 

### 3.2. Completion/Salvage Hysterectomy 

There is no demonstrated benefit of hysterectomy following brachytherapy in patients with LACC [76,77]. A randomized trial compared brachytherapy versus radical hysterectomy after external beam chemoradiation with gemcitabine plus cisplatin in patients with IB2-IIB cervical cancer. This study failed to demonstrate the superiority of surgery over the standard brachytherapy, with more severe complications in the surgical arm [78]. In a retrospective study, a post brachytherapy hysterectomy in locally advanced cervical cancer was prejudicial. The hysterectomy group had fewer local relapses, similar survival outcomes, and a higher incidence of severe late complication (22.5% vs. 6.5%). Urinary toxicity, such as a urinary fistula, was twice more frequent in the surgery group [79]. A retrospective series studied the treatment outcome of 29 patients who underwent a salvage hysterectomy when clinical and/or radiological residual disease was suspected [80]. Only 14 patients had histological residual disease, which revealed the low specificity of MRI and PET-CT imaging of less than 40% when identifying residual cervical cancers [80]. In addition, the dose escalation process offered by the integration of modern brachytherapy modalities into clinical routines led to local control rates of 95% in stage IB–IIB, and therefore, there is no longer a place for a complete hysterectomy in modern standards of care.

When there is a residual disease or relapse after chemoradiation plus brachytherapy, carefully selected patients (no disease extension outside the central pelvis, possibility to achieve a microscopically complete resection, acceptance of mutilating surgery) can benefit from salvage surgery. The prognosis of patients with a local relapse is very poor and the benefits of salvage surgery are questionable [81]. A single institute retrospective study reported that 10.8% of total patients treated for LACC between 2004 and 2016 had a local relapse [82]. Among these 28 patients, only 3 patients (10.7%) could be treated with salvage surgery. Two of them were still alive at 7.5 and 8 years [82]. Pelvic wall involvement, the possibility of permanent urinary derivation and colostomy, poor performance status, and the discovery of distant metastases before the surgery were the major limitations of salvage surgery, and were associated with high peri-operative morbidity and considerable quality of life impairment [82,83,84]. The poor outcome of salvage treatment reminds us of the importance of a meticulously executed curative upfront treatment at the stage of LACC. The prognosis of recurrent or metastatic cervical cancer remains dismal. Adding bevacizumab to cisplatin–paclitaxel increased the overall survival of patients with recurrent, persistent, or metastatic cervical cancer from 13.3 months to 17.0 months (HR for death, 0.71, 98% CI 0.54 to 0.95; *p* = 0.004). Contrary to chemotherapy, bevacizumab was also shown to be effective on target lesions that were located in the previously irradiated pelvis [85].

### 3.3. Neoadjuvant Chemotherapy 

Two large, randomized phase III trials (EORTC55994, NCT00039338) have evaluated the efficacy of neoadjuvant chemotherapy prior to a radical hysterectomy, both in patients with stage IB2-IIB cervical cancers. These trials have shown inferior outcomes with neoadjuvant chemotherapy, compared with the standard treatment based on upfront chemoradiotherapy followed by a brachytherapy boost. Both studies reported similar 5-year OS [86,87]. The first study reported a lower 5-year DFS in the neoadjuvant chemotherapy before radical hysterectomy group, compared with concomitant chemoradiation (paclitaxel plus carboplatin, HR, 1.38; 95% CI, 1.20 to 1.87) [86]. In the EORTC 55,994 study, only 76% of patients who received a cumulative minimal of 225 mg/m^2^ could undergo a radical hysterectomy, mainly due to disease progression under chemotherapy (24% of patients), an insufficient response to neo-adjuvant chemotherapy (16%), and treatment toxicities (34% of patients). In addition, one-third of the patients treated with neoadjuvant chemotherapy and surgery still needed post-operative radiotherapy, thereby increasing the risk of treatment-related complications [87]. 

Induction chemotherapy prior to concomitant chemoradiotherapy was compared with conventional concomitant chemoradiotherapy in a randomized phase II trial in patients with FIGO IIB to IVA cancers. Patients in the induction chemotherapy group had a lower 3-year PFS (40.9% vs. 60.4%), a lower 3-year OS (60.7% vs. 86.8%), and a lower complete response rate upon completion of the initial treatment (56.3% vs. 80.3%) compared with the immediate concomitant chemoradiotherapy group. The delay before starting chemoradiotherapy was detrimental, possibly because disease progression might occur during neoadjuvant chemotherapy, or because induction chemotherapy compromised the delivery of concomitant chemoradiotherapy [88]. The ongoing randomized comparative phase III INTERLACE trial (NCT01566240) compares induction chemotherapy plus standard chemoradiation to chemoradiation alone among patients with LACC. The induction chemotherapy consists of 6 cycles of weekly paclitaxel (80 mg/m^2^) plus carboplatin (AUC2). A prior phase II study found a response rate of 68% in patients treated with the same regimen prior to chemoradiation with acceptable toxicity. During the chemoradiotherapy stage, 22% of patients discontinued concurrent chemotherapy during chemoradiotherapy, mainly due to the cumulative toxicities; however, the compliance to radiotherapy was high (98%), and the 3-year OS was 67% [89]. 

### 3.4. Adjuvant Chemotherapy 

With the development of image-guided brachytherapy, 5-year local control rates reached 89–90% in the overall population of LACC [60,90]. Distant failure has become the most frequent pattern of relapse, thus increasing the interest in adjuvant therapy to increase the control of systemic disease. Adjuvant chemotherapy was tested in phase III randomized trials to reduce disease relapse and distance metastases. One study published in 2011 compared concurrent cisplatin–gemcitabine and radiation followed by adjuvant cisplatin–gemcitabine with conventional concomitant chemoradiotherapy in 515 patients with stage IIB to IVA cervical cancers [91]. The 3-year PFS (74.4% vs. 65.0%, HR 0.68, 95% CI 0.49 to 0.95) and 3-year OS (HR 0.68; 95% CI 0.49 to 0.95) were significantly improved in favor of the adjuvant chemotherapy group. Toxicities were more frequent in the adjuvant chemotherapy group (86.5% vs. 46.3%, *p* < 0.001), including three deaths related to treatment toxicity. Two of the three deaths were due to hematologic toxicities [91]. More recently, phase III OUTBACK trial (NCT01414608) from the GOG group, including 900 patients with FIGO stage IB-IVA cervical cancers, tested a more usual chemotherapy protocol of four cycles of carboplatin–paclitaxel as an adjuvant chemotherapy after concomitant chemoradiotherapy. This study showed no difference in the 5-year OS (72% vs. 71%, HR, 0.91, 95% CI 0.70 to 1.18) [92]. This treatment strategy has therefore not become the standard practice because of excessive toxicity over time and a lack of survival benefit; however, the limitation of the OUTBACK trial was that patients requiring interstitial brachytherapy upfront, or those with para-aortic lymph node metastases, were not included.

### 3.5. Adjuvant Immunotherapy 

In 2021, the U.S Food and Drug Administration approved one immunotherapy and one antibody–drug conjugate for metastatic cervical cancers. In October 2021, pembrolizumab was approved as part of a first-line regimen for patients with persistent, recurrent, or metastatic cervical cancer. The phase III Keynote-826 randomized comparative study tested pembrolizumab in patients with tumors expressing PD-L1. The pembrolizumab arm yielded longer PFS (10.4 vs. 8.2 months when PD-L1 ≥ 1%, HR, 0.62; 95% CI, 0.50 to 0.77; *p* < 0.001), 10.4 vs. 8.1 months when PD-L1 ≥ 10%, HR, 0.58; 95% CI, 0.44 to 0.77; *p* < 0.001) and higher 2-year OS (53.0% vs. 41.7% when PD-L1 ≥ 1%, HR, 0.64; 95% CI, 0.50 to 0.81; *p* < 0.001, 40.4% vs. 50.4% when PD-L1 ≥ 10%, HR, 0.67; 95% CI, 0.54 to 0.84; *p* < 0.001) [93]. Cemiplimab is a monoclonal antibody targeting PD1 that is approved for treatment of locally advanced or metastatic cutaneous squamous-cell carcinoma [94,95], and advanced non-small cell lung cancer with ≥50% PD-L1 expression [96]. The phase III Empower-cervical 1/GOG-3016/ENGOT-CX9 study (NCT03257267) randomized 608 patients, regardless of PD-L1 expression level to cemiplimab or investigator choice chemotherapy, in recurrent or metastatic cervical cancer after prior treatment by bevacizumab and paclitaxel. In patients with squamous cell carcinoma, cemiplimab increased the median OS from 8.8 months to 11.1 months (HR 0.73; 95% CI 0.58 to 0.91), and from 7.0 months to 13.3 months (HR 0.56; 95% CI 0.36 to 0.85) in patients with adenocarcinoma compared with patients in the chemotherapy group. Treatment related adverse events were lower with cemiplimab versus chemotherapy (25% vs. 45% anaemia, 18% vs. 33% nausea, 16% vs. 23% vomiting); however, cemiplimib showed unfavorable results when combined with hypofractionated radiotherapy in treating patients with recurrent or metastatic cervical cancer. All the patients receiving cemiplimab plus radiotherapy discontinued treatment, mainly due to disease progression or recurrence [97].

Following the promising results of immunotherapy in the metastatic setting, the addition of durvalumab is being evaluated in the phase III CALLA study (NCT03830866). This trial will randomize 770 high-risk patients with locally advanced cervical cancer with lymph node-positive diseases (FIGO stage IB2-IIB node-positive and IIIA-IVA with all lymph node status), either to chemoradiotherapy alone or concomitant chemoradiotherapy followed by a 24-month maintenance treatment using durvalumab [98]. A recent press release suggested that the study failed to meet its primary objective, but definitive results are required to better understand how immunotherapy acts in combination with chemoradiation in patients with LACC. The phase II AtezoLACC trial (NCT03612791) is randomizing 189 patients to undergo conventional chemoradiotherapy with or without atezolizumab, followed by maintenance treatment. The timing of atezolizumab delivery is being tested in another phase I clinical trial (NCT03738228) including 40 patients with FIGO stage IIIC cervical cancer. This trial aims to determine whether the association of atezolizumab with concomitant chemoradiotherapy results in differential immune activation, which is determined by the clonal expansion of T cell receptor beta repertoires in peripheral blood on day 21 [99].

### 3.6. Perspectives 

The identification of high-risk patterns for locoregional relapse currently relies on histopathological findings, such as tumor depth infiltration, tumor size, presence of LVSI or histological subtype (adenocarcinoma versus squamous cell carcinoma). Genomic profiling studies of advanced cervical cancer are promising, with data confirming that PIK3CA, STK11, PTEN, and the PI3K/AKT/mTOR pathway are the most frequently dysregulated pathways. An *ERBB3* mutation and a high tumor mutational burden are associated with prolonged survival in patients treated with anti-PD1 immunotherapies, and therefore they are identified as being predictive biomarkers in advanced cervical cancer [100]. The integration of molecular characteristics into clinical research for cervical cancers may help better analyze the risk of relapse, and therefore, it may lead to a substantial improvement in patients’ stratification for locoregional or systemic intensification. Other perspectives include radiomic studies to guide the response to immunotherapy as a neoadjuvant, concomitant, or adjuvant therapy. An analysis of the patterns of tumor response after primary chemoradiotherapy may also be useful to better guide the dose escalation process at time of brachytherapy, according to intrinsic tumor radiosensitivity [101,102]. Finally, there are numerous promising therapeutic pathways for salvage treatment of patients with refractory/persistent disease. Those include anti-cancer immunotherapy treatments using a cancer-derived multiple epitope–peptide cocktail vaccination, with early phase clinical trials suggesting the feasibility and effectiveness of this approach [103]. 

## 4. Conclusions

Despite the benefits afforded by technological improvements in terms of morbidity sparing (IMRT), the combination of surgery and radiotherapy in early-stage cervical cancer increases the risk of treatment-related complications, and proper patient selection is a prerequisite for achieving a high cure rate with acceptable morbidity after upfront surgery. After testing different combinations of multi-modality treatment, the standard of care in LACC is still concurrent chemoradiotherapy followed by a 3D image-guided brachytherapy boost. Salvage hysterectomy is only feasible in a small proportion of patients with LACC. (Neo-)adjuvant treatment strategies for cervical cancer, according to the FIGO 2018 classification, are illustrated in Figure 1. Clinical trials of neo-adjuvant chemotherapy plus surgery in FIGO IB-IIB diseases demonstrated the detrimental treatment outcome of such a combination, compared with upfront chemoradiotherapy plus brachytherapy. The OUTBACK Trial in FIGO stage IB-IVA cervical cancers showed that adjuvant chemotherapy after concomitant chemoradiotherapy did not improve overall survival. The INTERLACE study is testing the induction chemotherapy by paclitaxel and carboplatin before standard chemoradiation. The success of immunotherapy is increasing, with a high level of evidence that patients’ outcomes are improved at the metastatic stage. Results from ongoing immunoradiotherapy trials are required, as distant failure has now become the most frequent modality of failure. 

## Figures and Tables

**Figure 1 cancers-14-02449-f001:**
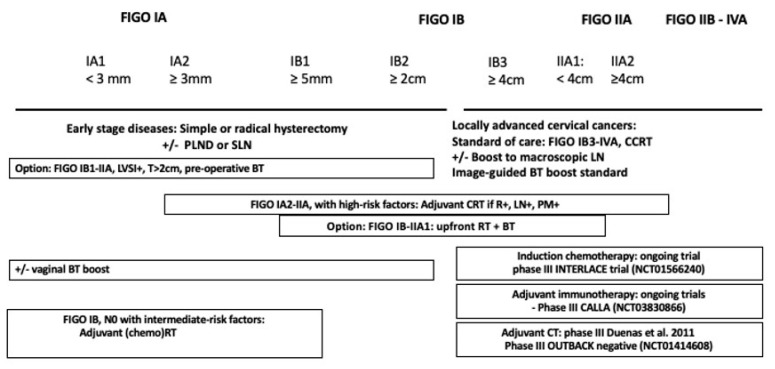
Treatment algorithm for cervical cancers according to FIGO 2018 classification [22,24,26,27,28,29,31,32,43,44,45,52,53,54,55,56,57,58,60,61,62,63,64,65,67,68,69,70,71,72,91]. FIGO, Féderation Internationale de Gynécologie et d’Obstétrique; PLND, pelvic lymph node dissection; SLN, sentinel lymph node; LN, lymph node; CCRT, concurrent chemoradiotherapy; RT, radiotherapy; BT, brachytheraph; LVSI, lymphovascular space invasion; DSI, Deep stromal invasion.

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
