# Peer review of "Current Standards in the Management of Early and Locally Advanced Cervical Cancer: Update on the Benefit of Neoadjuvant/Adjuvant Strategies"

_cancers, 2022, doi:10.3390/cancers14102449_

Round 1
Reviewer 1 Report
abstract:
p1 l20, there is an error, the main location of recurrence is at a distance
Introduction:
- I don't see the relevance of prevention measures in this review
- p1 l38, a reference on the clinical impact or an explanation would be interesting
- p2 l46-48, this sentence does not belong here but rather after the following sentence
- p2 l56, the adjuvant treatment is radiotherapy but also chemotherapy +/- brachytherapy
- p2 l60, the reference is very old
- p2 l61, the review also covers neoadjuvant treatments
Second part
- p2 l78, results should be provided with 95% CIs
- p2 l83, again the reference is old, it dates from 1990
- p3 l115, a reference should be put
same remark for the following sentence (l116-118) - p4 l192, 2cm and not 2mm
- p5 l245-246, 95% CIs are missing, are the results significant?
Part three
- p8 l359-362, ICs are still missing
Author Response
Dear Reviwer,
Thank you for the excellent suggestions. We have made the changes according to your instructions. Please see the attachment.
Reviewer 2 Report
Line 31 WHO – capitalize each word
Line 54 modalities of surgical interventions – laparotomy, laparoscopy, robotic surgery
Line 153 positive margins instead of positive marge
Line 290 please specify highly selected patients
Line 334 adjuvant therapy typo mystake
Line 388 insert reference
Recommendations:
- association of immunotherapy with radiotherapy and chemoradiotherapy
for cervical cancer
- treatment of bulky pelvic lymph nodes in cervical cancer
- genomic profiling (PIK3CA, PTEN, ERBB3, and PI3K/AKT pathway, TMB),
and predictive biomarkers in cervical cancer.
- cancer-derived multiple epitope-peptides cocktail vaccination in
refractory/persistent disease of cervical cancer
- the therapeutic strategies on the association between pregnancy and
cervical cancer
Author Response
Line 54 modalities of surgical interventions – laparotomy, laparoscopy, robotic surgery
We thank the review for the suggestions. This subject is a separate topic. We didn’t discuss it in details after deliberation. The context of fertility sparing approaches and of cervical cancer occurring in pregnancy are excluded from this literature review, as both issues were recently addressed extensively.
Line 153 positive margins instead of positive marge
We have now corrected the language error. Thank you very much.
They had FIGO IA2-IIA cervical cancer who had positive pelvic lymph nodes and/or positive margins and/or microscopic involvement of parametrium after upfront radical hysterectomy and PLND
Line 290 please specify highly selected patients
It is an excellent suggestion. We specified the criteria for highly selected patients.
When there is a residual disease or relapse after chemoradiation plus brachytherapy, highly selected patients (no disease extension outside the central pelvis, possibility to perform a complete resection, acceptance of mutilating surgery) can benefit from salvage surgery.
Line 334 adjuvant therapy typo mystake
We thank the reviewer for the maticulous proof reading. We have corrected this error.
Distant failure has become the most frequent pattern of relapse thus increasing the interest in adjuvant therapy to increase the control of systemic disease.
Line 388 insert reference
Indeed, a reference is very much needed for this study. We have inserted the protocol for the Atezolizumab study.
This trial aim to determine whether the association of atezolizumab with concomitant chemoradiotherapy results in differential immune activation, determined by clonal expansion of T cell receptor beta repertoires in peripheral blood on day 21 [100]
Mayadev, J., D. Zamarin, W. Deng, H. Lankes, R. O'Cearbhaill, C.A. Aghajanian, and R. Schilder, Anti-PD-L1 (atezolizumab) as an immune primer and concurrently with extended-field chemoradiotherapy for node-positive locally advanced cervical cancer. Int J Gynecol Cancer, 2020. 30(5): p. 701-704.
Suggestion: genomic profiling (PIK3CA, PTEN, ERBB3, and PI3K/AKT pathway, TMB), and predictive biomarkers in cervical cancer.
Genomic profiling studies of advanced cervical cancer confirmed that PIK3CA, STK11, PTEN, and the PI3K/AKT/mTOR pathway to be the most frequently dysregulated pathways. ERBB3 mutation and high tumour mutational burden were associated with prolonged survival in patients treated with anti-PD1 immunotherapies, and therefore identified as predictive biomarkers in advanced cervical cancer [101]. We are looking forward to seeing the substantial treatment improvement put forward by the active research in cervical cancers.
Suggestions:
- Cancer-derived multiple epitope-peptides cocktail vaccination in refractory/persistent disease of cervical cancer
This strategy was addressed in the revised manuscript
- The therapeutic strategies on the association between pregnancy and cervical cancer
Reviewer 3 Report
The authors present an overview of the current standards in the management of early and locally advanced cervical cancer. The article is well organized and clinically clear. In section 3.2 authors should mention the role of bevacizumab in the treatment of persistent and local relapse.
Duvalumab? -Durvalumab line 377
Author Response
The authors present an overview of the current standards in the management of early and locally advanced cervical cancer. The article is well organized and clinically clear. In section 3.2 authors should mention the role of bevacizumab in the treatment of persistent and local relapse.
We thank the reviwer so very much for the kind comment.
Yes, we have mentioned the role of bevacizumab in the treatment of cervical cancer:
Adding bevacizumab to cisplatin-paclitaxel increased the overall survival patients with recurrent, persistent, or metastatic cervical cancer from 13.3 months to 17.0 months (HR for death, 0.71, 98% CI 0.54 to 0.95; p = 0.004). Contrary to chemotherapy, bevacizumab was also shown effective on target lesions that were located in the previously irradiated pelvis (86).
Duvalumab? -Durvalumab line 377
We thank the reviwer for correcting our error.
Following the promising results of immunotherapy in the metastatic setting, the addition of durvalumab is being evaluated in the phase III CALLA study (NCT03830866).